# Anti-Inflammatory Effects of Barley Sprout Fermented by Lactic Acid Bacteria in RAW264.7 Macrophages and Caco-2 Cells

**DOI:** 10.3390/foods13111781

**Published:** 2024-06-06

**Authors:** Sang-Hyun Kim, Youn Young Shim, Young Jun Kim, Martin J. T. Reaney, Mi Ja Chung

**Affiliations:** 1College of Veterinary Medicine and Institute of Animal Medicine, Gyeongsang National University, Jinju 52828, Republic of Korea; 2Department of Food and Bioproduct Sciences, University of Saskatchewan, Saskatoon, SK S7N 5A8, Canada; younyoung.shim@usask.ca (Y.Y.S.); martin.reaney@usask.ca (M.J.T.R.); 3Prairie Tide Diversified Inc., Saskatoon, SK S7J 0R1, Canada; 4Department of Food and Biotechnology, Korea University, Sejong 30019, Republic of Korea; yk46@korea.ac.kr; 5Department of Food Science and Nutrition, College of Health Welfare, Gwangju University, Gwangju 61743, Republic of Korea

**Keywords:** barley sprout, lactic acid bacteria, *Lactiplantibacillus plantarum*, *Latilactobacillus curvatus*, Caco-2 cells, anti-inflammatory effects

## Abstract

The anti-inflammatory effects of supernatants produced from sprouted barley inoculated with *Lactiplantibacillus plantarum* KCTC3104 (Lp), *Leuconostoc mesenteroides* KCTC3530 (Lm), *Latilactobacillus curvatus* KCTC3767 (Lc), or a mixture of these lactic acid bacteria were investigated using RAW264.7 macrophages. BLp and BLc, the lyophilized supernatants of fermented sprouted barley inoculated with Lp and Lc, respectively, effectively reduced the nitric oxide (NO) levels hypersecreted by lipopolysaccharide (LPS)-stimulated RAW264.7 and LPS-stimulated Caco-2 cells. BLp and BLc effectively reduced the NO levels in LPS-stimulated RAW264.7 macrophages, and these effects tended to be concentration-dependent. BLc and BLp also exhibited strong DPPH radical scavenging activity and immunostimulatory effects. BLp and BLc significantly suppressed the levels of NO and pro-inflammatory cytokines such as TNF-α, IL-1β, and IL-6 in LPS-stimulated RAW264.7 macrophages and LPS-stimulated Caco-2 cells, indicating their anti-inflammatory effects. These effects were greater than those of unfermented barley sprout (Bs). The functional components of Bs, BLp, and BLc were analyzed by HPLC, and it was found that lutonarin and saponarin were significantly increased in the fermented sprouted barley sample inoculated with Lp and Lc (BLp and BLc).

## 1. Introduction

In recent years, the rise in inflammatory diseases has raised concerns worldwide. Factors such as gut microbiota dysbiosis, genetic predisposition, and environmental triggers contribute to the escalating incidence of chronic inflammation [1]. In this context, lactic acid bacteria (LAB) have emerged as promising candidates for the treatment of inflammation due to their antimicrobial properties and widespread applications in the food and pharmaceutical industries [2].

Barley (*Hordeum vulgare* L.) sprouts, which are rich in bioactive compounds, have attracted attention for their potential health benefits such as anti-obesity effects, anti-inflammatory effects, antioxidative activity, and preventive effects against alcohol-induced liver injury [3,4,5,6]. Barley has high concentrations of dietary fibers, minerals, flavonoids, and phenolic acids [7,8,9]. Among the known phenolic acids and flavonoids found in barley are saponarins and lutonarins [8,9]. Researchers have explored their anti-inflammatory effects, particularly when subjected to fermentation with LAB [10,11]. One such strain, *Lactiplantibacillus* (*Lp.*) *plantarum*, has shown remarkable stability in withstanding variations in temperature and pH, making it an attractive candidate for modulating inflammation [12]. *Latilactobacillus* (*Ll.*) *curvatus* can alleviate colitis by inhibiting colonic inflammation on dextran sulfate sodium (DSS)-induced Caco-2 cell damage [13]. The RAW264.7 macrophage and Caco-2 cell line models in this study allow researchers to study the effects of functional materials on immune responses and gut health [2,13,14]. RAW264.7 macrophages and Caco-2 cells were treated with LPS to be used in in vitro pro-inflammatory cell models [15,16].

LAB, including *Lp. plantarum* and *Ll. curvatus*, produce extracellular vesicles (ECVs) with potent anti-inflammatory effects [14]. These ECVs inhibit the production of pro-inflammatory cytokines, such as tumor necrosis factor-α (TNF-α) and interleukin-6 (IL-6), in macrophages [14]. By modulating these cytokines, LAB-derived ECVs play a critical role in regulating immune responses. LAB-fermented foods and beverages may contain ECVs secreted by LAB, such as *Lp. plantarum* and *Ll. curvatus* [14].

Fermented foods and beverages have been widely used in the last century due to the health benefits of increasingly bioactive components produced by various probiotic microorganisms in these foods [17]. Generally, fermented dairy products and kimchi are known to be the main source of probiotic microorganisms in Republic of Korea [18], and *Lp. plantarum* and *Ll. curvatus* are abundant in kimchi [19]. Prebiotics are non-digestible food ingredients that benefit the host by selectively stimulating the growth or activity of one or a limited number of bacteria in the colon [19]. Beta-glucans have been used as prebiotic fiber [11,20], and barley contains beta-glucans [21]. Barley sprouts are used as prebiotics that support *Lp. plantarum* and *Ll. curvatus* [15].

Probiotic (or LAB)-derived cell vesicles (extracellular vesicles, ECVs) are nanoparticle-scale structures with beneficial postbiotic effects. Fermentation-derived plant functional compounds contain ECVs that are delivered to the epithelial cells of the intestinal mucosa and, potentially, to other tissues [22]. Barley sprouts fermented by LAB may contain ECVs. In this study, the purpose of fermenting barley sprouts with LAB such as *Lp. plantarum*, *Leuconostoc* (*Le.*) *mesenteroides*, and *Ll. curvatus* was as follows. First, barley sprouts, which are known to induce anti-inflammatory effects, are rich in dietary fiber, such as beta-glucans, and can be used as food for LAB [21]. Second, in our previous study, ECVs secreted by *Lp. plantarum*, *Le. mesenteroides*, and *Ll. curvatus*, respectively, had immunostimulatory effects and anti-inflammatory effects in RAW264.7 macrophages [14], so barley sprout fermented by *Lp. plantarum*, *Le. mesenteroides*, and *Ll. curvatus* will add to the anti-inflammatory effects of ECVs. Thus, it was hypothesized that barley sprout fermented by *Lp. plantarum*, *Le. mesenteroides*, and *Ll. curvatus* would induce greater anti-inflammatory effects than barley sprout that was not fermented. Third, the process of fermentation of barley sprouts may increase the functional components [8,9] that show anti-inflammatory effects in barley sprouts, or it may produce new metabolites that affect the anti-inflammatory activity of the extracts.

The present study investigated the potential of barley sprouts fermented by LAB to reduce the inflammatory effects in lipopolysaccharide (LPS)-stimulated RAW264.7 murine macrophages and LPS-stimulated human intestinal Caco-2 cells in vitro. To correlate the anti-inflammatory compounds produced by fermentation with bioactivity, high-performance liquid chromatography was used to analyze the active compounds in unfermented and fermented barley sprouts.

## 2. Materials and Methods

### 2.1. Barley and Culture of LABs

The sprouted barley used in this study was certified as organically grown in Gochang (Jeollabuk-do, Republic of Korea). The sprouted barley was harvested when it grew to approximately 20 cm, dried, ground, sterilized at 121 °C for 15 min, and cooled before being used for fermentation. The *Lp. plantarum* KCTC3104 (Lp), *Leuconostoc mesenteroides* KCTC3530 (Lm), and *Ll. curvatus* KCTC3767 (Lc) strains were obtained from the Korean Collection for Type Cultures of the Korea Research Institute of Bioscience and Biotechnology (Daejeon, Republic of Korea).

All LAB strains were cultured overnight in MRS broth (#288130; Difco, Sparks, MD, USA). All LAB counts were measured using the method described by Kim et al. (2017a, 2017b) [23,24]. For bacterial enumeration, the optical density (OD) of the bacterial cultures was adjusted (A600 = 0.5). Tenfold serial dilutions of the OD-adjusted cultures were prepared with PBS, and then 0.1 mL of each of the diluted bacterial suspensions were plated onto MRS agar plates in triplicate and incubated at 37 °C for 24 h. The resulting bacterial counts, recorded as log colony-forming units (CFU) per milliliter, were used to calculate the Lp, Lm, and Lc numbers required for the preparation of barley sprout with LAB.

### 2.2. Sprouted Barley Sample Preparation and Fermentation

Samples for LAB inoculation were prepared as shown in Table 1, and 50 mL conical tubes were inoculated with Lp, Lm, and Lc. For the preparation of F (fermented) samples, 7 g of sprouted barley powder was mixed with 35 mL of drinking water and inoculated with Lp, Lm, and Lc (F-Lp, F-Lm, F-Lc). In addition, 7 g of sprouted barley powder was mixed with 35 mL of drinking water and then inoculated with Lp and Lm for F-Lp+Lm; Lp and Lc for F-Lp+Lc; Lm and Lc for F-Lm+Lc; and Lp, Lm, and Lc for F-Lp+Lm+Lc. Three replicates of each sample were prepared. Prior to fermentation, samples were mixed with a vortex mixer (WiseMix VM-10, DAIHAN Scientific Co., Wonju, Republic of Korea) for 1 min. The final number of LAB inoculated in all the samples was 10^4^ colony-forming units (CFU)/g. LAB-inoculated samples were incubated at 37 °C for 3 days. All experimental samples were free of LAB prior to inoculation.

### 2.3. Sample Preparation for Cell Treatment and Antioxidative Activity, and HPLC Analyses

LAB counts were determined in sprouted barley fermentations inoculated with LAB and incubated for 3 days. *Lactobacilli* MRS agar plates (BD Difco, Sparks, MD, USA) were used for LAB colony counting, and the number of colonies formed after 48 h of incubation at 37 °C were determined in duplicate, averaged, and expressed as CFU/g. To obtain samples for treating RAW264.7 macrophages and Caco-2 to investigate antioxidative and immunomodulatory effects, the fermented samples were centrifuged (4000× *g*, 15 min) to obtain the supernatant and then lyophilized. Freeze-dried samples were labeled with sample label B (barley sprout) instead of sample label F (fermented) in Table 1. F-Lm was used to distinguish between Lm-inoculated sprouted barley fermentations and the supernatants of these fermentations, and lyophilized samples were labeled BLm (fermented barley sprout with Lm). Samples for the 2,2-diphenyl-1-picrylhydrazyl (DPPH) radical scavenging activity determination and HPLC analysis are the same as those for cell treatment.

### 2.4. Determination of DPPH Radical Scavenging Activity

The DPPH (Sigma-Aldrich Co., St. Louis, MO, USA) free-radical scavenging activity was determined by mixing the sample (540 μL) at certain concentrations (1, 5, 10 mg/mL) with DPPH (1.5 × 10^−4^ M, 360 μL) solution, reacting at 37 °C for 30 min, and then measuring the absorbance at 517 nm [25]. The DPPH radical scavenging activity was expressed as a % by [1 − (absorbance of sample/absorbance of control)] × 100 by comparing the absorbance of the added sample to that of the control [26].

### 2.5. Cell Line Culture

RAW264.7 cells, a mouse macrophage line, and Caco-2 cells derived from the human colon were obtained from the Korean Cell Line Bank (KCLB, Seoul, Republic of Korea). RAW264.7 cells were cultured in Dulbecco’s Modified Eagle Medium (DMEM; WelGene Co., Daegu, Republic of Korea) containing 10% fetal bovine serum (FBS; WelGene Co.) solution and 1% penicillin and streptomycin (PEST; WelGene Co.) solution, and Caco-2 cells were cultured in Minimum Essential Medium (MEM; WelGene Co.) containing 10% FBS and 1% PEST (WelGene Co.). The cells were cultured in an incubator (BB 15, Thermo Electron LED GmbH, Langenselbold, Germany) at 37 °C, 5% CO_2_.

### 2.6. Measurement of Cell Viability and Determination of Nitric Oxide (NO) and Cytokines

To determine cell viability, cells were seeded at 1 × 10^4^ cells/well into 96-well plates and incubated for 24 h before being replaced with FBS-free serum-free medium (SFM) and incubated for an additional 2 h. After fermentation of the sprouted barley mixture compositions listed in Table 1, lyophilized supernatants were prepared and used to treat cells at concentrations ranging from 50 to 250 μg/mL for each sample to determine their effects on cell viability. RAW264.7 macrophages were cultured for 24 h, and Caco-2 cells were cultured for 3 days. Cell viability was measured using the 3-(4,5-dimethylthiazol-2-yl)-2,5-dipheylterazolium bromide (MTT, Sigma-Aldrich Co.) reduction method [27]. Twenty-four hours after sample treatment, the media were replaced with 5 mg/mL MTT solution mixed with SFM medium in a 1:10 ratio. After 4 h in an incubator at 37 °C and 5% CO_2_, the medium was carefully removed, leaving formazan, produced by the reduction of MTT. The remaining medium was removed by incubation at room temperature for 30 min in the absence of light, and then the samples were dissolved using dimethyl sulfoxide (DMSO; Sigma-Aldrich Co.), and the absorbance was measured at 570 nm with DMSO as blank. Cell viability (%) was calculated as the absorbance of the sample treatment group divided by the absorbance of the control group and multiplied by 100.

For NO and cytokine production measurements, RAW264.7 cells were seeded at 5 × 10^5^ cells/well into 24-well plates and incubated for 48 h, then replaced with SFM medium and incubated for another 2 h. Samples with supernatant added at various concentrations were treated for 18 h. Samples treated with LPS (0.01 μg/mL, Sigma-Aldrich. Co.) were treated instead of these samples as a positive control to investigate the immunostimulatory effect. To investigate the anti-inflammatory effect, samples pretreated for 3 h were subsequently treated with LPS (0.01 μg/mL) for 18 h.

Caco-2 cells were seeded at 5 × 10^5^ cells/well in 24-well plates and cultured for 7 days with medium changes every 2–3 days, then replaced with SFM medium and cultured for another 2 h. To investigate the anti-inflammatory effect, samples were treated with LPS (1 μg/mL) for 18 h after 3 h of treatment with the specified supernatant.

The amount of NO produced by RAW264.7 cells was measured by adding 0.5 mL of Griess reagent (1% sulfanilamide in 5% phosphoric acid and 1% *N*-(1-naphthyl)ethylenediamine in H_2_O) to 0.5 mL of the cell culture medium, mixing, and allowing the mixture to stand at room temperature for 10 min after light blocking, and then measuring the absorbance at 540 nm [14]. The NO content was calculated by constructing a standard curve with sodium nitrate. TNF-α, IL-1β, and IL-6 levels in the cell cultures were measured using ELISA kits (eBioscience Co., San Diego, CA, USA) according to the manufacturer’s instructions.

### 2.7. HPLC Analysis of Lutonarin and Saponarin

High-performance liquid chromatography (HPLC) was performed on a Perkin Elmer Flexar, Quat™ system with a pump, autosampler, and photodiode array detector (Perkin Elmer, Shelton, CT, USA) according to the method of Uy et al. (2023) [28]. Samples were quantitatively analyzed on a reversed-phase HPLC system using a YMC Pack-pro™ C18 column (250 mm × 4.6 mm, 5 μm). The column temperature was maintained at 30 °C. The mobile phase consisted of 0.2% acetic acid in water (A) and acetonitrile (ACN) (B). The analysis was performed using gradient elution conditions of 50% A: 50% B after 30 min in an eluent composition of 10% A water and 90% B. The flow rate was 1 mL/min, and 10 μL injections were used. The detector wavelength was 330 nm.

### 2.8. Statistical Analysis

Data were presented as mean ± standard deviation (SD), and the average values were derived from three to eight values per experiment. Each experiment was repeated at least three times. All data were analyzed by the one-way analysis of variance (ANOVA) using IBM SPSS for Windows version 18.0 software (SPSS Inc., Chicago, IL, USA). The differences in results between treatments were confirmed by Duncan’s multiple-range test. Statistical significance was accepted at *p* < 0.05.

## 3. Results and Discussion

### 3.1. DPPH Radical Scavenger Activity and Anti-Inflammatory Effects of Fermented Barley Sprout with LAB in LPS-Stimulated RAW264.7 Cells

BLc (85.4%) exhibited higher DPPH radical scavenger activity than other samples at 10 mg/mL (Figure 1A). Unfermented barley sprout (Bs) and fermented barley sprout samples, which were fermented by Lm, Lp+Lm and Lp+Lc (BLm, BLp+Lm, BLp+Lc) had strong DPPH radical scavenger activity at 5 mg/mL, but there was no significant change between fermented barley sprout and unfermented barley sprout. All samples had more than 75% DPPH radical scavenger activity at 10 mg/mL. Bs, BLp, BLc and BLp+Lc had no cytotoxicity at 50–250 μg/mL in RAW264.7 macrophages (Figure 1B). Therefore, the treatment concentrations used in further experiments on RAW264.7 macrophages were up to 250 μg/mL.

Bs, BLp, BLm, BLc, BLp+Lm, BLp+Lc, BLm+Lc, and BLp+Lm+Lc, respectively, were used at a concentration of 250 μg/mL to pretreat LPS-stimulated macrophages. The results of the inhibitory effects on NO production are shown in Figure 1C. The excessive secretion of NO in LPS-stimulated RAW264.7 macrophages was significantly reduced by supernatants from barley sprout and LAB-fermented barley sprout. The BLp and BLc treatment groups were more effective than the BLm group, and the BLp and the BLc treatment groups inhibited NO production in LPS-stimulated RAW264.7 macrophages more than other samples did. Therefore, BLp and BLc were selected, and further studies were conducted with Bs, BLp, BLc, and BLp+Lc. Bs, BLp, BLc, and BLp+Lc reduced the level of NO in LPS-induced RAW264.7 macrophages in a dose-dependent manner (Figure 2A). BLp, BLc, and BLp+Lc reduced NO levels more significantly than Bs in LPS-stimulated RAW264.7 cells, but there was no significant difference between BLp, BLc, and BLp+Lc at 100 μg/mL. There was no significant difference in NO production in LPS-stimulated RAW264.7 cells between the fermented barley sprout inoculated with Lp and Lc and the fermented barley sprout inoculated with two LABs (Lp+Lc) (Figure 2A).

The concentrations of TNF-α, IL-1β, and IL-6 were also highly elevated after cells were stimulated with LPS. Bs, BLp, and BLc treatments reduced TNF-α, IL-1β, and IL-6 levels in LPS-stimulated RAW264.7 macrophages (Figure 2B–D) in a concentration-dependent manner. BLp and BLc treatments reduced TNF-α and IL-6 levels in LPS-stimulated RAW264.7 cells more significantly than the Bs treatment. This was similar to the NO response to these treatments. TNF-α production was most effectively suppressed by BLc treatment at 100 μg/mL in LPS-stimulated RAW264.7 macrophages (Figure 2B). There was no effect of supernatants prepared after inoculation with two types of LABs (Lp+Lc) compared with inoculation with a single type of LAB (Lp, Lc) on the TNF-α and IL-1β suppression in LPS-stimulated RAW264.7 macrophages (Figure 2B,C), and this was also similar to the trend in the NO accumulation results.

The DPPH radical scavenging activity of the ethanol extract of barley sprouts at 1000 μg/mL was 92%, and the 70% ethanol extract of barley sprouts also significantly suppressed the LPS-induced production of NO and TNF-α [29]. The 75% ethanol extract of barley sprouts showed stronger antioxidative activity than other extracts [30]. The DPPH radical scavenging activity of the supernatant of Lp and Lc fermented barley sprouts was lower than the ethanol extracts of non-fermented barley sprouts, but their anti-inflammatory activity was similar (Figure 1A). Fermented barley sprouts contained various bioactivity compounds [28]. During fermentation, LAB produce a range of secondary metabolites, some of which have been associated with health-promoting properties [31]. The functional ingredients contained in BLp and BLc are metabolites and water-soluble compounds. The supernatant of fermented barley sprouts also had 80–90% potent DPPH radical scavenging activity. The compounds produced during barley sprout fermentation are more effective as anti-inflammatories than as antioxidants.

In our previous study, supernatants prepared from barley sprouts fermented with Lm increased the production of NO, TNF-α, IL-1β, and IL-6 in RAW264.7 cells [15]. Therefore, we suggested that BLm is an immunostimulatory material. However, the anti-inflammatory effects of Lm- fermented barley sprouts have not been reported. In this study, the anti-inflammatory effects of BLm were tested, and it was shown that BLp and BLc reduced NO production in LPS-stimulated RAW264.7 cells (Figure 1C) and LPS-stimulated Caco-2 cells (Figure 3B) more effectively than BLm. Therefore, Lm was excluded from further experiments.

This study compared the anti-inflammatory effects of unfermented barley sprouts and barley sprouts fermented by LAB. Most cell types release ECVs into the extracellular space. LAB medium contains ECVs, and BLp and BLc are the lyophilized supernatants of LP- or Lc-inoculated sprouted barley ferment, so BLp and BLc will also contain ECVs from kimchi-derived LAB, such as Lc and Lp, which modulate immune responses [14]. ECVs contained in BLp and BLc are associated with anti-inflammatory effects, so it is expected that BLp and BLc treatments will have more anti-inflammatory effects than Bs treatment in LPS-stimulated RAW264.7 cells. The following experiments were conducted to determine whether the anti-inflammatory effects of the treatments on RAW264.7 cells could be replicated in human intestinal Caco-2 cells.

### 3.2. Effects of Fermented Barley Sprout with LAB on NO and TNF-α Production in LPS-Stimulated Caco-2 Cells

Treatments with Bs, BLp, BLc, and BLp+Lc were not cytotoxic to Caco-2 cells at 250 μg/mL (Figure 3A), and therefore treatments with Bs, BLp, BLc, and BLp+Lc were conducted with up to 250 μg/mL in Caco-2 cells.

Pretreatment of cells with Bs, BLp, BLm, BLc, BLp+Lm, BLp+Lc, or BLp+Lm+Lc (250 μg/mL) inhibited LPS-induced NO production in Caco-2 cells (Figure 3B). The NO secretion by Caco-2 cells measured after pretreatment with BLp, BLc or BLp+Lc and subsequent treatment with LPS was the same as that produced in LPS-free controls. The BLp, BLc, and BLp+Lc treatments were more effective in suppressing NO than Bs, BLm, BLp+Lm, BLm+Lc, and BLp+Lm+Lc in LPS-stimulated Caco-2 cells (Figure 3B).

Treatment of Caco-2 cells with LPS in the absence of Bs, BLp, BLm, BLc, BLp+Lm, BLp+Lc, BLm+Lc, and BLp+Lm+Lc, respectively, inhibited LPS-induced NO production, but there was no significant difference between the effects of supernatants from unfermented barley sprouts (Bs) and supernatants from fermented barley sprouts (BLp, BLm, BLc, BLp+Lm, BLp+Lc, BLm+Lc, BLp+Lm+Lc) (Figure 3C). Subsequently, Bs, BLp, and BLc were selectively used for the pretreatment before the LPS treatment in Caco-2 cells.

Treatment with BLp and BLc significantly inhibited LPS-induced NO and TNF-α production in a dose-dependent manner (Figure 4). BLp inhibited TNF-α secretion significantly more than BLc and Bs at 100 μg/mL in LPS-stimulated Caco-2 cells (Figure 4B). Other plant extracts showed similar effects. For example, the treatment of Caco-2 cells with LPS increased NO production, whereas an extract of *Taraxacum coreanum* Nakai significantly decreased NO production and TNF-α production compared with the LPS-only control [16]. Tu et al. (2016) [32] suggested that chitosan nanoparticles suppressed the LPS-induced inflammatory response in Caco-2 cells by decreasing the secretion of pro-inflammatory cytokines. Our study shows increased NO and TNF-α production in Caco-2 cells after LPS exposure. However, treatment with BLp and BLc attenuated these LPS-induced inflammatory responses by decreasing NO and TNF-α production in LPS-stimulated Caco-2 cells. The barley sprout extracts prepared after fermentation with LAB were more effective at inhibiting NO and TNF-α production than the unfermented barley sprout in LPS-stimulated Caco-2 cells. This was similar to the results in the LPS-stimulated RAW264.7 macrophages, and the reason for the results was that the fermented barley sprouts were more effective in suppressing NO and TNF-α production than the unfermented barley sprouts in LPS-stimulated RAW264.7 macrophages. In other words, the barley sprouts fermented by Lp and Lc may contain ECVs derived from these LAB as well as increased bioactive compounds of barley sprouts, and these will have anti-inflammatory effects. Our results suggest that BLp and BLc have anti-inflammatory effects in LPS-stimulated Caco-2 cells and RAW264.7 cells, which may be potential therapeutic agents against intestinal inflammation [16].

### 3.3. Immunostimulatory Effects of Fermented Barley Sprout with LAB in RAW264.7 Cells

To evaluate the immune-enhancing activity of Bs, BLp, BLm, and BLp+Lm, the levels of NO, TNF-α, IL-1β, and IL-6 secreted by RAW264.7 cells were examined. NO and TNF-α levels in RAW264.7 cells were significantly increased compared with those in untreated cells (control) (Figure 5A,B). NO and TNF-α levels were significantly higher in BLp or BLc-treated cells than in Bs-treated cells at the same concentration. In particular, BLc and BLp+Lc showed the highest NO and TNF-α levels, but there was no significant difference between BLc and BLp+Lc at 100 μg/mL (Figure 3A,B). BLc increased TNF-α levels more effectively than BLp+Lc at 25 and 50 μg/mL. LPS (0.01 μg/mL) was used as a positive control, and LPS treatment significantly increased IL-6 (1173.7 ± 13.2 pg/mL) and IL-1β (164.9 ± 3.0 pg/mL) levels compared with the control and sample groups in RAW264.7 cells. When RAW264.7 cells were stimulated with 100 μg/mL Bs, BLp, BLc, and BLp+Lc, respectively, the production of IL-1β and IL-6 increased more than in the untreated control group (Figure 5C,D). There was no change in the production of IL-1β and IL-6 between the control group and those treated with fermented barley sprout extracts (Bs, BLp, BLc, and BLp+Lc at 25 μg/mL and 50 μg/mL; Figure 3C,D). The production of TNF-α and IL-6 was greater in the BLc-treated group than in the BLp-treated group (Figure 5B,D).

The second aim of this work was to evaluate the immunostimulatory properties of fermented barley sprouts by Lp and Lc. Hot-water extracts of barley sprouts increased the production of NO, TNF-α, IL-6, and IL-1β in RAW264.7 cells [33]. Kim et al. (2022) [34] fermented the hot-water extract of barley sprouts with *Lp. plantarum* KCL005 and *Le. mesenteroides* KCL007 to confirm the immunostimulatory effects in RAW264.7 cells. When the barley sprout fermentation product was used to treat RAW264.7 cells, increased production of NO, TNF-α, IL-1β, and IL-6 was observed [27]. The NO, TNF-α, IL-1β, and IL-6 levels were increased in macrophages because plant-derived polysaccharides such as β-glucan were increased in the barley fermented with LAB [35,36]. Our previous studies have demonstrated that ECVs of *Ll. Curvatus* and *Lp. plantarum* significantly increase NO, TNF-α, IL-1β, and IL-6 levels in RAW264.7 cells [11]. In this study, NO and cytokine levels in BLp- and BLc-treated cells may be increased due to the increase in exopolysaccharides, β-glucan, metabolites, and ECVs due to LAB fermentation [14,35,36].

### 3.4. Anti-Inflammatory Compound Analysis Using HPLC

Two of the major anti-inflammatory compounds in fermented barley sprout, lutonarin and saponarin, were quantified by HPLC. The chromatograms obtained from fermented barley sprouts and standard references are shown in Figure 6, showing that lutonarin was the most abundant, followed by saponarin (Table 2). The concentration of lutonarin and saponarin increased more in BLp and BLc than in Bs (Table 2).

The flavonoids lutonarin and saponarin are major components of fermented barley sprouts, and lutonarin suppressed the LPS-induced upregulation of IL-6 and TNF-α through suppression of the NF-kB signaling pathway in LPS-stimulated RAW264.7 macrophages [37]. Saponarin from barley sprouts downregulated the expression of the pro-inflammatory mediator IL-6 and inhibited the NF-κB and MAPK pathways in LPS-induced RAW264.7 cells [38]. Thus, lutonarin and saponarin have been reported as promising natural anti-inflammatory agents [37,38]. The major compounds of ethanol extracts of barley sprouts are rutin, gallic acid, ferulic acid, and ρ-coumaric acid [30]. These extracts also have antioxidative and anti-inflammatory activities [29,30]. Uy et al. (2023) [28] reported that lutonarin, saponarin, isoorientin, isovitexin, and tricin were detected in fermented barley sprouts.

The higher anti-inflammatory effect of BLp and BLc than Bs in LPS-stimulated RAW264.7 cells and LPS-stimulated Caco-2 cells might be due to the increase in lutonarin and saponarin in BLp and BLc. Further studies should be conducted to examine the changes in lutonarin and saponarin during the fermentation of Lp- and Lc-inoculated sprouted barley, the fermentation conditions under which these substances are most abundant, and the bioactive compounds and metabolites that change during the fermentation of sprouted barley. Untargeted metabolomics is needed in future studies to reveal the metabolites in fermented barley sprout.

## 4. Conclusions

These results indicate that the anti-inflammatory effects of unfermented barley sprout (Bs) and fermented barley sprout (BLp and BLc) are due to the decrease in NO and pro-inflammatory cytokines in LPS-stimulated RAW264.7 macrophages and LPS-stimulated human intestinal Caco-2 cells. Furthermore, BLp and BLc treatment significantly enhanced the anti-inflammatory effects by increasing lutonarin and saponarin production during barley sprout fermentation. Therefore, we suggest that BLp and BLc should be considered potential candidates for anti-inflammatory agents for treating intestinal inflammation-related diseases. Understanding the mechanisms underlying these interactions may bring us closer to new strategies for managing inflammation and improving overall well-being.

## Figures and Tables

**Figure 1 foods-13-01781-f001:**
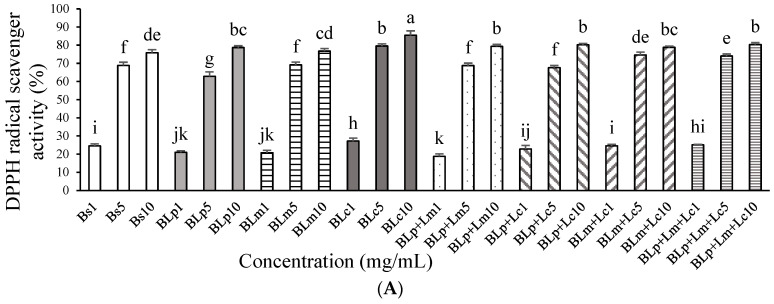
Effect of unfermented barley sprouts and fermented barley sprouts on (**A**) DPPH radical scavenger activity, (**B**) cell viability of RAW264.7 macrophage cells, and (**C**) NO production of LPS-stimulated RAW264.7 macrophage cells. Bs, BLp, BLm, BLc, BLp+Lm, BLp+Lc and BLp+Lm+Lc are the lyophilized samples of the supernatant of the unfermented and fermented barley sprouts shown in Table 1. The viability was measured by the MTT assay, and the RAW264.7 cells were treated with the sample for 24 h. (**C**) The RAW264.7 cells were treated with LPS for 18 h after pretreatment of the sample for 3 h. Values are expressed as the mean ± SD (*n* = 3), and those followed by different letters (a–k) within a property are significantly different (*p* < 0.05) according to Duncan’s multiple-range test. n.s.; not significantly different from each other.

**Figure 2 foods-13-01781-f002:**
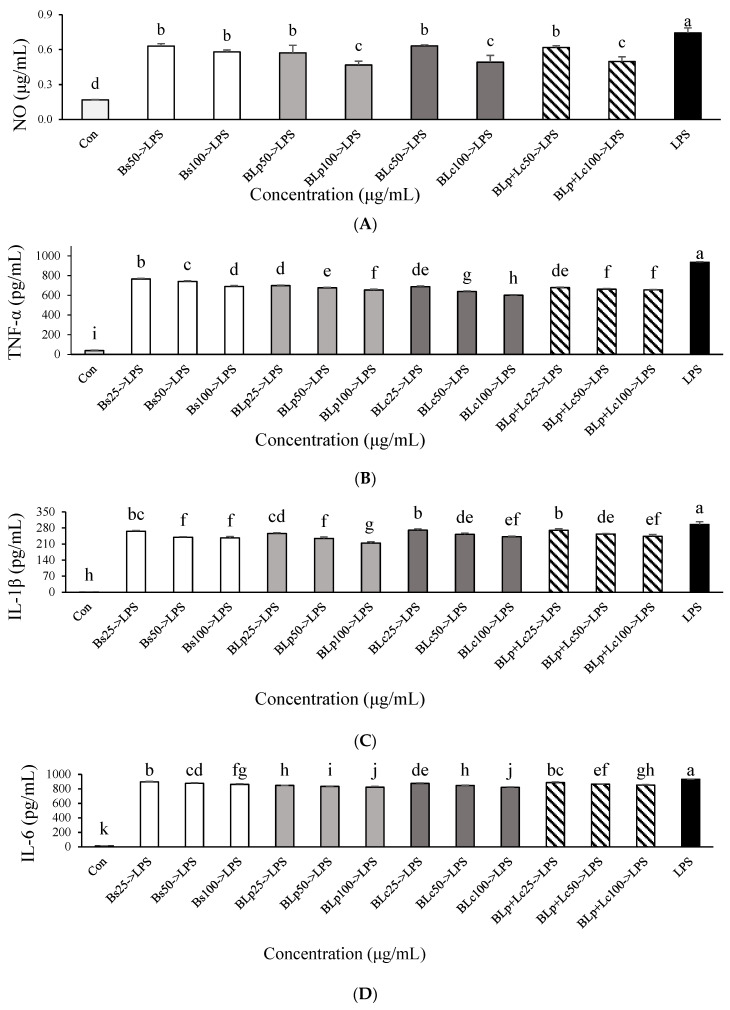
Effect of unfermented barley sprouts and fermented barley sprouts on the production of (**A**) NO, (**B**) TNF-α, (**C**) IL-1β, and (**D**) IL-6 in LPS-stimulated RAW264.7 macrophage cells. Bs, BLp, BLc, and BLp+Lm are lyophilized samples of the supernatant of the unfermented and fermented barley sprouts shown in Table 1. The RAW264.7 cells were treated with LPS for 18 h after pretreatment of the sample for 3 h. Values are expressed as the mean ± SD (*n* = 3), and those followed by different letters (a–k) within a property are significantly different (*p* < 0.05) according to Duncan’s multiple-range test.

**Figure 3 foods-13-01781-f003:**
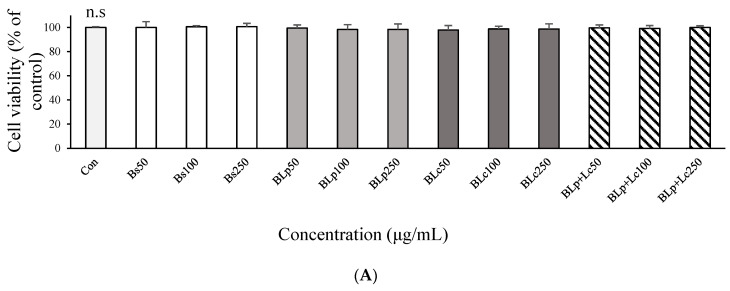
Effect of unfermented and fermented barley sprouts on the (**A**) cell viability of Caco-2 cells and (**B**,**C**) NO production of LPS-stimulated Caco-2 cells. Bs, BLp, BLm, BLc, BLp+Lm, BLp+Lc, and BLp+Lm+Lc are the lyophilized dry samples of the supernatant of the unfermented and fermented barley sprouts shown in Table 1. (**A**) The viability was measured by the MTT assay and the Caco-2 cells were treated with the sample for 24 h. (**B**) The Caco-2 cells were treated with LPS for 18 h after pretreatment of the sample for 3 h. (**C**) The Caco-2 cells were co-treated with the samples in the presence or absence of LPS for 18 h. Values are expressed as the mean ± SD (*n* = 3), and those followed by different letters (a–e) within a property are significantly different (*p* < 0.05) according to Duncan’s multiple-range test. n.s.; not significantly different from each other.

**Figure 4 foods-13-01781-f004:**
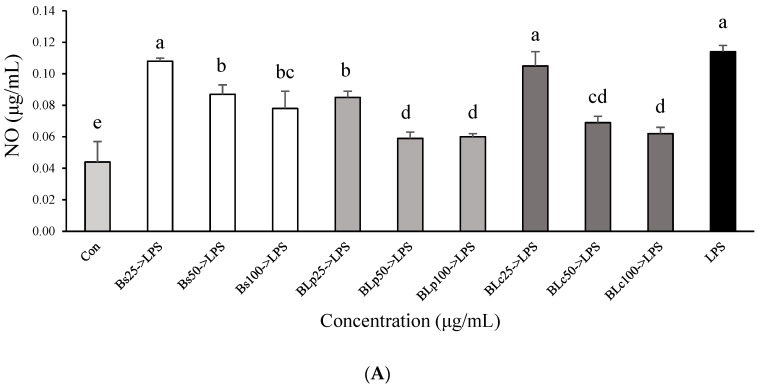
Effect of unfermented and fermented barley sprouts on the production of (**A**) NO and (**B**) TNF-α in LPS-stimulated Caco-2 cells. Bs, BLp, BLc, and BLp+Lm are lyophilized dry samples of the supernatant of the unfermented and fermented barley sprouts shown in Table 1. The Caco-2 cells were treated with LPS for 18 h after pretreatment of the sample for 3 h. Values are expressed as the mean ± SD (*n* = 3), and those followed by different letters (a–f) within a property are significantly different (*p* < 0.05) according to Duncan’s multiple-range test.

**Figure 5 foods-13-01781-f005:**
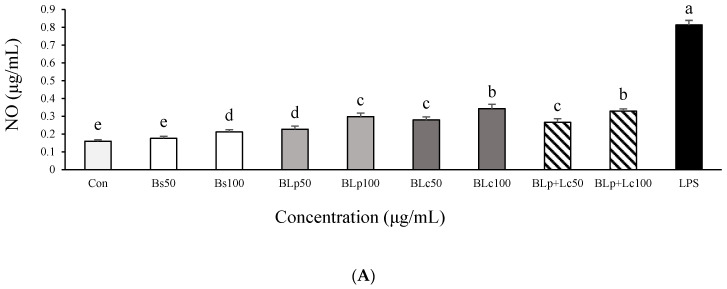
Effect of unfermented and fermented barley sprouts on the production of (**A**) NO, (**B**) TNF-α, (**C**) IL-1β, and (**D**) IL-6 in RAW264.7 macrophage cells. Bs, BLp, BLc, and BLp+Lm are the lyophilized dry samples of the supernatant of the unfermented and fermented barley sprouts shown in Table 1. The RAW264.7 cells were treated with the sample for 18 h. LPS (0.01 μg/mL) was used as a positive control. Values are expressed as the mean ± SD (*n* = 3), and those followed by different letters (a–l) within a property are significantly different (*p* < 0.05) according to Duncan’s multiple-range test.

**Figure 6 foods-13-01781-f006:**
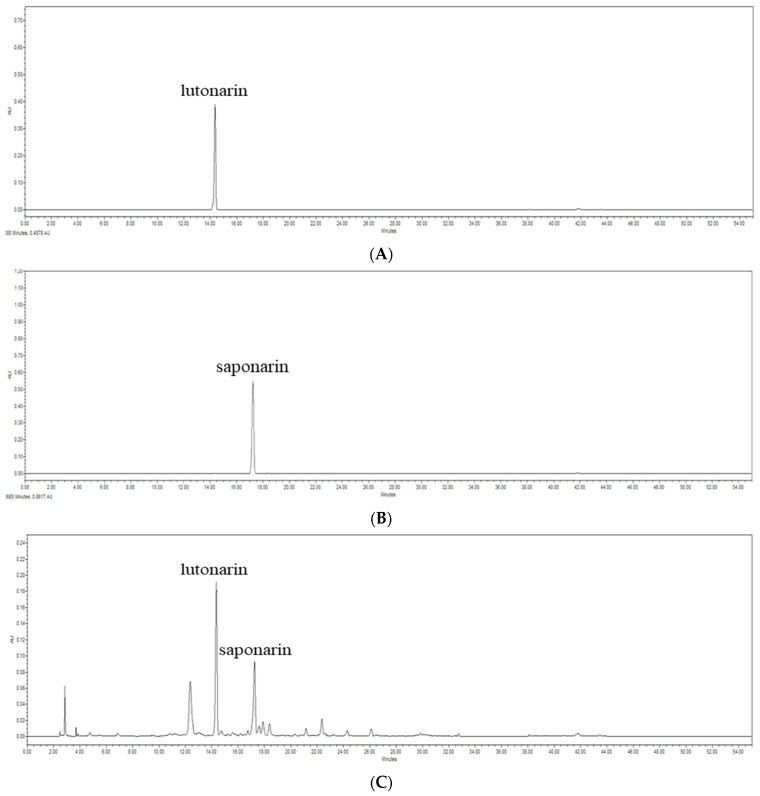
Chromatograms from the HPLC analysis of standards ((**A**) lutonarin and (**B**) saponarin) and (**C**) fermented barley sprouts.

**Table 1 foods-13-01781-t001:** Composition of barley sprout mixtures.

Group ^1^	Ingredient				
	Barley Sprout Powder (g)	Water (mL)	Lp	Lm	Lc
F-Bs	7	35			
F-Lp	7	35	10^4^ CFU/g		
F-Lm	7	35		10^4^ CFU/g	
F-Lc	7	35			10^4^ CFU/g
F-Lp+Lm	7	35	10^4^ CFU/g	10^4^ CFU/g	
F-Lp+Lc	7	35	10^4^ CFU/g		10^4^ CFU/g
F-Lm+Lc	7	35		10^4^ CFU/g	10^4^ CFU/g
F-Lp+Lm+Lc	7	35	10^4^ CFU/g	10^4^ CFU/g	10^4^ CFU/g

^1^ F: fermented; Bs: Barley sprout without LAB; Lp: *Lp. plantarum* KCTC3104; Lm: *Le. mesenteroides* KCTC3530; Lc: *Ll. curvatus* KCTC3767.

**Table 2 foods-13-01781-t002:** Lutonarin and saponarin content in the fermented sprout barley.

Samples ^1^	Concentration (mg/g)	
	Lutonarin	Saponarin
Bs	4.92 ± 0.05 ^c^	2.17 ± 0.01 ^c^
BLp	5.49 ± 0.05 ^a^	2.44 ± 0.04 ^b^
BLc	5.23 ± 0.06 ^b^	2.57 ± 0.01 ^a^

^1^ Bs: Barley sprouts without LAB; BLp: the fermented barley sprouts with *L. plantarum*; BLc: the fermented barley sprouts with *Ll. curvatus*. The samples were then fermented with the barley sprout mixtures shown in Table 1. The values are expressed as mean ± SD (*n* = 3); means with different letters (a–c) in the same column significantly differ from each other (*p* < 0.05), as determined by Duncan’s multiple-range test.

## Data Availability

The original contributions presented in the study are included in the article, further inquiries can be directed to the corresponding author.

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
