# Peer review of "Anti-Inflammatory Effects of Barley Sprout Fermented by Lactic Acid Bacteria in RAW264.7 Macrophages and Caco-2 Cells"

_foods, 2024, doi:10.3390/foods13111781_

Round 1
Reviewer 1 Report
Comments and Suggestions for Authors
In the introduction, please describe the idea of fermenting barley sprouts, is it merely to obtain the inflammatory effects or for human consumption.
Line 40: “health benefits”.. such as??
Line 74: why organic barley?
It seems that the extraction of barley sprout was missing in materials and method
Line 77: what is the reason using these particular strains? Does these strains possesses several physiological effects from the previous study?.
Line 89: do you have any reason why the colony number is 104 cfu instead of 109/108 cfu?.
The discussion can be improved
It would be nice if the authors provide the figures of RAW cells to see the viability
Is there any metabolites other than lutonarin and saponarin that may exhibit anti-inflammatory effect.
Untargeted metabolomic is needed in future study to reveal the metabolites in fermented barley sprout
Author Response
Thank you for your patience and recommendations for strengthening our manuscript (ID: foods -3013670). We would like to thank the reviewers for their time and expertise in providing critical feedback to make this manuscript suitable for publication. We have revised our manuscript according to the editors’ and reviewers’ comments. In addition to these changes, we have also made substantial revisions to improve this manuscript's style, flow, and clarity. We hope these changes improve the overall quality of this manuscript for publication. We have listed the reviewers’ comments and answered them in sequence.
Reviewer #1:
Comment: In the introduction, please describe the idea of fermenting barley sprouts, is it merely to obtain the inflammatory effects or for human consumption.
Response: The concept of fermenting barley sprouts was described in the introduction on Lines 69-85.
Probiotic (or LAB)-derived cell vesicles (extracellular vesicles, ECVs) are nanoparticle-scale structures with beneficial postbiotic effects. Fermentation-derived plant functional compounds contain ECVs that are delivered to the epithelial cells of the intestinal mucosa and potentially to other tissues [22]. Barley sprouts fermented by LAB may contain ECVs. In this study, the purpose of fermenting barley sprouts with LAB such as Lp. plantarum, Leuconostoc (Le.) mesenteroides, and Ll. curvatus were as follows. First, barley sprouts, which are known to induce anti-inflammatory effects, are rich in dietary fiber such as beta-glucans and can be used as food for LAB [21]. Second, in our previous study, ECVs secreted by Lp. plantarum, Le. mesenteroides, and Ll. curvatus, respectively, had immunostimulatory effects and anti-inflammatory effects in RAW264.7 macrophages [14], so barley sprout fermented by Lp. plantarum, Le. mesenteroides, and Ll. curvatus will add to the anti-inflammatory effects of ECVs. Thus, it was hypothesized that barley sprout fermented by Lp. plantarum, Le. mesenteroides, and Ll. curvatus would induce greater anti-inflammatory effects than barley sprout that was not fermented. Third, the process of fermentation of barley sprouts may increase the functional components [8,9] that show anti-inflammatory effects in barley sprouts, or may produce new metabolites that affect the anti-inflammatory activity of the extracts.
- Liu, Y.; Alexeeva, S.; Defourny, K.; Smid, E.; Abee, T. Tiny but mighty: bacterial membrane vesicles in food biotechnological applications. Curr. Opin. Biotechnol. 2018, 49, 179–184.
Comment: “health benefits”.. such as?
Response: We have added the sentences in the Introduction section on Lines 40-42.
Barley (Hordeum vulgare L.) sprouts, which are rich in bioactive compounds, have attracted attention for their potential health benefits such as anti-obesity effects, anti-inflammatory effects, antioxidant activity, and preventive effects against alcohol-induced liver injury [3–6].
- Obadi, M.; Sun, J.; Xu, B. Highland barley: Chemical composition, bioactive compounds, health effects, and applications. Food Res. Int. 2021, 140, 110065.
- Lee, Y.H.; Kim, J.H.; Kim, S.H.; Oh, J.Y.; Seo, W.D.; Kim, K.M.; Jung, J.C.; Jung, Y.S. Barley sprouts extract attenuates alcoholic fatty liver injury in mice by reducing inflammatory response. Nutrients 2016, 8, 440.
- Kim, M.J.; Kawk, H.W.; Kim, S.H.; Lee, H.J.; Seo, J.W.; Kim, J.T.; Jang, S.H.; Kim, M.J.; Kim, Y.M. Anti-obesity effect of hot water extract of barley sprout through the inhibition of adipocyte differentiation and growth. Metabolites 2021,11(9), 610.
- Chae, K.S.; Ryu, E.H.; Kim, K.D.; Kim, Y.S.; Kwon, J.W. Antioxidant activities of ethanol extracts from barley sprouts. Korean J. Food Sci. Technol. 2019, 51, 486-491.
Comment: why organic barley?
Response: Barley sprout powder products sold in Korea are mainly organic barley. Therefore, organic barley was used in this study.
Comment: It seems that the extraction of barley sprout was missing in materials and method.
Response: After fermentation, the sample was centrifuged. The supernatant was frozen and used in the experiment. The barley sprouts were not extracted.
Comment: Line 77: what is the reason using these particular strains? Does these strains possesses several physiological effects from the previous study?
Response: In our previous study, ECVs secreted by Lp, Lm, and Lc, respectively had the immunostimulatory effects and anti-inflammatory effects in RAW264.7 macrophages [14], and so barley sprout fermented by Lp, Lm, and Lc will add the anti-inflammatory effects of ECVs. Thus, barley sprout fermented by Lp, Lm, and Lc were expected to have a better anti-inflammatory effect than the anti-inflammatory effects of barley sprouts.
- Kim, S.H.; Lee, J.H.; Kim, E.H.; Reaney, M.J.T.; Shim, Y.Y.; Chung, M.J. Immunomodulatory activity of extracellular vesicles of kimchi-derived lactic acid bacteria (Leuconostoc mesenteroides, Latilactobacillus curvatus, and Lactiplantibacillus plantarum). Foods 2022, 11(3), 313.
Comment: Line 89: do you have any reason why the colony number is 104 cfu instead of 109/108 cfu?
Response: To inoculate 109/108 cfu lactic acid bacteria, it is necessary to cultivate large amounts of lactic acid bacteria. In the preliminary experiment, it was well fermented even when inoculated with 104 cfu of lactic acid bacteria. Therefore, it is better to inoculate 104cfu lactic acid bacteria for commercial use.
Comment: The discussion can be improved.
Response: We improved the discussion.
Comment: It would be nice if the authors provide the figures of RAW cells to see the viability
Response: In Figure 1B, we have showed the cell viability of raw264.7 macrophages by MTT assay.
Comment: Is there any metabolites other than lutonarin and saponarin that may exhibit anti-inflammatory effect.
Response: We have added the sentences in Results and Discussion section on lines 449-451.
Uy et al. (2023) reported that lutonarin, saponarin, isoorientin, isovitexin, and tricin were detected in fermented barley sprouts.
- Uy, N.P.; Lee, H.D.; Byun, D.C.; Lee, S. Comparison of the contents of total polyphenol, total flavonoid, and flavonoids derivatives in unfermented and fermented barley sprouts. J App Bio Chem. 2023. 66, 353–358.
Comment: Untargeted metabolomic is needed in future study to reveal the metabolites in fermented barley sprout
Response: We have added the sentences in Results and Discussion section on Lines 454, 459.

Reviewer 2 Report
Comments and Suggestions for Authors
This manuscript titled “Anti-inflammatory Effects of Barley Sprout Fermented by Lactic Acid Bacteria in RAW264.7 Macrophages and Caco-2 Cells” (No. foods-3013670) investigated the anti-inflammatory effects of unfermented barley sprout and fermented barley sprout based on the decrease of NO and proinflammatory cytokines in LPS-stimulated RAW264.7. This work provides some valuable insights to lactic acid bacteria-fermented barley malt which not only increased the saponin active compounds, but also improved the utilization value of barley malt in anti-inflammation. The experiments designed for this study were well performed with a lot of data. However, some drawbacks still exist in the present version of this manuscript, and thus authors have to make a major revision to their manuscript before resubmitting to the journal Foods. Specific points are issued as follows.
1/Line 17-18 “FD-Lp (lyophilized supernatant of Lp-inoculated sprouted barley ferment) and FD-Lc”, and line 19-20, “BLp (lyophilized supernatant of Lp-inoculated sprouted barley ferment) and BLc”, which is right?
2/Line 18, “LPS” indicates what?
3/What is relation between line 57-65 and this study?
4/Line 67-68, “lipopolysaccharide (LPS)-stimulated RAW264.7 murine macrophages and LPS-stimulated human intestinal Caco-2 cells in vitro”, why use LPS-stimulated RAW264.7 murine macrophages and human intestinal Caco-2 cells?
5/For section 2.1, how to activate the LAB strains?
6/Line 81, how much inoculum of bacterial cells?
7/Line 113-115, how to make the samples for determination of DPPH?
8/Line 163-165, how to make samples for HPLC test?
9/Line 221-224, which results could support this conclusion?
10/What is relation between line 228-234 and line 265-272?
11/Line 275-277 and line 232-235, they are repeatedly.
12/For section 3.2, discussion about the effects of fermented barley sprout with LAB on NO and TNF-α production in LPS-stimulated Caco-2 cells should be added.
13/Line 317-319, and line 373-375, references are needed.
14/Line 367-369, which data could favor these results?
15/Line 402-404, which data display these compounds, such as rutin, gallic acid, ferulic acid, and ρ-coumaric acid of barley sprouts?
16/For Line 405-407, a little bit discussion should be added.
17/For part references, journal name should be unified. And the Latin names for species should be written in italic (for instance, see line 457-458).
Author Response
Anti-inflammatory Effects of Barley Sprout Fermented by Lactic Acid Bacteria in RAW264.7 Macrophages and Caco-2 Cells
Thank you for your patience and recommendations for strengthening our manuscript (ID: foods -3013670). We would like to thank the reviewers for their time and expertise in providing critical feedback to make this manuscript suitable for publication. We have revised our manuscript according to the editors’ and reviewers’ comments. In addition to these changes, we have also made substantial revisions to improve this manuscript's style, flow, and clarity. We hope these changes improve the overall quality of this manuscript for publication. We have listed the reviewers’ comments and answered them in sequence.
Reviewer #2:
This manuscript titled “Anti-inflammatory Effects of Barley Sprout Fermented by Lactic Acid Bacteria in RAW264.7 Macrophages and Caco-2 Cells” (No. foods-3013670) investigated the anti-inflammatory effects of unfermented barley sprout and fermented barley sprout based on the decrease of NO and proinflammatory cytokines in LPS-stimulated RAW264.7. This work provides some valuable insights to lactic acid bacteria-fermented barley malt which not only increased the saponin active compounds, but also improved the utilization value of barley malt in anti-inflammation. The experiments designed for this study were well performed with a lot of data. However, some drawbacks still exist in the present version of this manuscript, and thus authors have to make a major revision to their manuscript before resubmitting to the journal Foods. Specific points are issued as follows.
Comment: 1/Line 17-18 “FD-Lp (lyophilized supernatant of Lp-inoculated sprouted barley ferment) and FD-Lc”, and line 19-20, “BLp (lyophilized supernatant of Lp-inoculated sprouted barley ferment) and BLc”, which is right?
Response: We have modified the following sentence on lines 18-19.
BLp and BLc, lyophilized supernatant of sprouted barley ferment inoculated with Lp and Lc, respectively, effectively reduced the nitric oxide (NO) levels hypersecreted by lipopolysaccharide (LPS)-stimulated Raw264.7 and LPS-stimulated Caco-2 cells.
Comment: 2/Line 18, “LPS” indicates what?
Response: Inserted the full name of lipopolysaccharide (LPS) on Line 20.
Comment: 3/What is relation between line 57-65 and this study?
Response: We explained the benefits of fermented foods and why we used lactic acid bacteria from kimchi, a popular Korean fermented food, in this study. We also explained why we chose sprouted barley to inoculate with these kimchi lactic acid bacteria to make fermented foods.
Comment: 4/Line 67-68, “lipopolysaccharide (LPS)-stimulated RAW264.7 murine macrophages and LPS-stimulated human intestinal Caco-2 cells in vitro”, why use LPS-stimulated RAW264.7 murine macrophages and human intestinal Caco-2 cells?
Response: The reasons for using RAW264.7 macrophages and Caco-2 cells have already been described in sections 50-53 of the Introduction.
" RAW264.7 macrophage and Caco-2 cell line models in this study allow researchers to study the effects of functional materials on immune responses and gut health [2,13,14]].”
The sentence below has been added. RAW264.7 macrophages and Caco-2 cells were treated with LPS to use in vitro pro-inflammatory cell models [15,16].]
Comment: 5/For section 2.1, how to activate the LAB strains?
6/Line 81, how much inoculum of bacterial cells?
Response: On lines 80-87, we have added the following sentences.
All LAB strains were cultured overnight in MRS broth (#288130; Difco, Sparks, MD, USA). All LAB counts were measured using the method described by Kim et al. (2017a, 2017b [23,24]. For bacterial enumeration, the optical density (OD) of the bacterial cultures was adjusted (A600 = 0.5). Tenfold serial dilutions of the OD-adjusted cultures were prepared with PBS and then 0.1 mL of each of the diluted bacterial suspensions were plated onto MRS agar plates, in triplicate, and incubated at 37 °C for 24 h. The resulting bacterial counts, recorded as log colony-forming units (CFU) per milliliter, were used to calculate the Lp, Lm and Lc numbers required for the preparation of barley sprout with LAB.
- Kim, S.H.; Kang, K.H.; Kim,S.H., Lee, S.; Lee, S.H.; Ha, E.S.; Sung, N.J.; Kim, J.G.; Chung, M.J. Lactic acid bacteria directly degrade N-nitrosodimethylamine and increase the nitrite-scavenging ability in kimchi. Food Control. 2017, 71, 101-109.
- Kim, S.H.; Kim,S.H. Kang, K.H.; Lee, S.; Kim, S.J.; Kim, J.G.; Chung, M.J. Kimchi probiotic bacteria contribute to reduced amount of N-nitrosodimathylamine in lactic acid bacteria-fortified kimchi. LWT-Food Science and Technology. 2017, 84, 196-203.
Comment: 7/Line 113-115, how to make the samples for determination of DPPH?
8/Line 163-165, how to make samples for HPLC test?
Response: Line 124 and, Lines 135,136
The following sentences were changed in the revised manuscript.
2.3. Sample Preparation for Cell Treatment, Antioxidant Activity and HPLC Analysis
Samples for DPPH radical scavenging activity and HPLC analysis are the same as those for cell treatment.
Comment: 9/Line 221-224, which results could support this conclusion?
Response: We improved the discussion on Lines 250-254.
Figure 1A). Fermented barley sprouts contained various bioactivity compounds [28]. During fermentation, LAB produce a range of secondary metabolites, some of which have been associated with health-promoting properties [31]. The functional ingredients contained in BLp and BLc will be metabolites and water-soluble compounds.
- Stanton, C.; Ross, R.P.; Fitzgerald, G.F.; Sinderen, D.V. Fermented functional foods based on probiotics and their biogenic metabolites. Current Opinion in Biotechnology, 2005, 16, 198-203.
Comment: 10/What is relation between line 228-234 and line 265-272?
Response: We improved the discussion on Lines 257-274.
In our previous study, supernatants prepared from barley sprouts fermented with Lm increased the production of NO, TNF-α, IL-1β, and IL-6 in RAW264.7 cells [15]. Therefore, we suggested that BLm is an immunostimulatory material. However, the anti-inflammatory effects Lm- fermented barley sprouts have not been reported. In this study, the anti-inflammatory effects of BLm were tested and the BLp and BLc reduced NO production in LPS-stimulated RAW264.7 cells (Figure 1C) and LPS-stimulated Caco-2 cells (Figure 3B) more effectively than the BLm. Therefore, Lm was excluded from continued further experiments.
This study compared the anti-inflammatory effects of unfermented barley sprouts and barley sprouts fermented by LAB. Most cell types release ECVs into the extracellular space. LAB medium contains ECVs, BLp and BLc were lyophilized supernatant of LP- or Lc-inoculated sprouted barley ferment and so BLp and BLc will also contain ECVs from Kimchi-derived LAB, such as Lc and Lp, which modulate immune responses [14]. ECVs contained in BLp and BLc are associated with anti-inflammatory effects, so it is expected that BLp and BLc treatments will have more anti-inflammatory effects than Bs treatment in LPS-stimulated RAW264.7 cells. The following experiments were conducted to determine whether the anti-inflammatory effects of the treatments on RAW264.7 cells could be replicated in human intestinal Caco-2 cells.
Comment: 11/Line 275-277 and line 232-235, they are repeatedly.
Response: Both sentences were deleted on Lines 317.
Lines 275-277
Lp and Lc were added to the supernatant of unfermented barley and fermented barley sprouts to investigate their anti-inflammatory effects on LPS-stimulated human intestinal Caco-2 cells.
Lines 232-235
BLp and BLc were investigated to identify compounds associated with anti-inflammatory effects.
Comment: 12/For section 3.2, discussion about the effects of fermented barley sprout with LAB on NO and TNF-α production in LPS-stimulated Caco-2 cells should be added.
Response: On Lines 357-365, the discussion about the effects of fermented barley sprout with LAB on NO and TNF-α production in LPS-stimulated Caco-2 cells added.
The barley sprout extracts prepared after fermentation with LAB were more effective at inhibition of NO and TNF-α production than the unfermented barley sprout in LPS-stimulated Caco-2 cells. This was similar to the results in the LPS-stimulated RAW264.7 macrophages and the reason for the results was that the fermented barley sprouts were more effective in suppressing NO and TNF-α production than the unfermented barley sprouts in LPS-stimulated RAW264.7 macrophages. In other words, the barley sprouts fermented by Lp and Lc may contain ECVs derived from these LAB and increased bioactive compounds of barley sprouts and they will have anti-inflammatory effects.
Comment: 13/Line 317-319, and line 373-375, references are needed.
Response: The references added [14,16,35,36] on Lines 367, 420.
Our results suggest that BLp and BLc have anti-inflammatory effects in LPS-stimulated Caco-2 cells and RAW264.7 cells, which may be potential therapeutic agents against intestinal inflammation [16].
]. In this study, NO and cytokine levels in BLp and BLc-treated cells may be increased due to the increase in exopolysaccharides, β-glucan, metabolites and ECVs due to LAB fermentation [14,35,36].
Comment: 14/Line 367-369, which data could favor these results?
Response: We deleted the sentence.
Plant-derived polysaccharides, such as β-glucan, arabinoxylan show immunostimulatory effects through the pattern receptors of macrophages.
Comment: 15/Line 402-404, which data display these compounds, such as rutin, gallic acid, ferulic acid, and ρ-coumaric acid of barley sprouts?
Response: Lines 449, 450: Water-soluble bioactive substances are present in the samples used in this study because the fermented and unfermented samples were prepared after adding water to the sprouted barley. Rutin, gallic acid, ferulic acid, and ρ-coumaric acid are the bioactive substances in the ethanol extract.
The major compounds of ethanol extracts of barley sprouts were rutin, gallic acid, ferulic acid, and ρ-coumaric acid [30].
Comment: 16/For Line 405-407, a little bit discussion should be added.
Response: Lines 452-459, we have added the sentences in Results and Discussion section.
The higher anti-inflammatory effect of BLp and BLc than Bs in LPS-stimulated RAW264.7 cells and LPS-stimulated Caco-2 cells might be due to the increase of lutonarin and saponarin in BLp and BLc. Further studies should be conducted on the changes in lutonarin and saponarin during fermentation of Lp and Lc inoculated sprouted barley, the fermentation conditions under which these substances are most abundant, and the bioactive compounds and metabolites that change during fermentation of sprouted barley. Untargeted metabolomic is needed in future study to reveal the metabolites in fermented barley sprout.
Comment: 17/For part references, journal name should be unified. And the Latin names for species should be written in italic (for instance, see line 457-458).
Response: Fixed on Lines 515,516

Round 2
Reviewer 1 Report
Comments and Suggestions for Authors
The authors have improved the manuscript
Reviewer 2 Report
Comments and Suggestions for Authors
Authors have revised their manuscript, and no more comments.